# Regional adaptations and parallel mutations in Feline panleukopenia virus strains from China revealed by nearly-full length genome analysis

**Élcio Leal** [1,2☯]*, **Ruiying Liang** [1,3☯], **Qi Liu** [1], **Fabiola Villanova** [2], **Lijun Shi** [1,3], **Lin Liang** [1,3], **Jinxiang Li** [1]*, **Steven S. Witkin** [4,5], **Shangjin Cui** [1,3]*

**1** Chinese Academy of Agricultural Sciences, Institute of Animal Sciences, Beijing, China, **2** Federal University of Pará, Belém, Pará, Brazil, **3** Beijing Observation Station for Veterinary Drug and Veterinary Biotechnology, Ministry of Agriculture, Beijing, China, **4** Department of Obstetrics and Gynecology, Weill Cornell Medicine, New York, NY, United States of America, **5** Institute of Tropical Medicine, Sao Paulo, Brazil

☯ These authors contributed equally to this work.

\* lijinxiang@caas.cn (JL); cuishangjin@caas.cn (SC); elcioleal@gmail.com (EL)

**Data Availability Statement:** FPV sequences were deposited in the Genbank Strain LSJ-2014 was deposited in Genbank with the number: MH165481) and CSJ-2015 MH165482.

## Abstract

Protoparvoviruses, widespread among cats and wild animals, are responsible for leukopenia. Feline panleukopenia virus (FPLV) in domestic cats is genetically diverse and some strains may differ from those used for vaccination. The presence of FPLV in two domestic cats from Hebei Province in China was identified by polymerase chain reaction. Samples from these animals were used to isolate FPLV strains in CRFK cells for genome sequencing. Phylogenetic analysis was performed to compare our isolates with available sequences of FPLV, mink parvovirus (MEV) and canine parvovirus (CPV). The isolated strains were closely related to strains of FPLV/MEV isolated in the 1960s. Our analysis also revealed that the evolutionary history of FPLV and MEV is characterized by local adaptations in the *Vp2* gene. Thus, it is likely that new FPLV strains are emerging to evade the anti-FPLV immune response.

## 1. Introduction

Carnivore protoparvovirus 1 is a member of the family *Parvoviridae* (subfamily *Parvovirinae*, genus *Protoparvovirus*) that includes several closely related viruses such as feline panleukopenia virus (FPLV), the mink enteritis virus (MEV) and canine parvoviruses (CPV) [1–10]. Protoparvoviruses are non-enveloped, small icosahedral viruses having a single-stranded DNA genome of approximately 5 Kb. The expression pattern of these viruses have two open reading frames (ORFs) encoding the non-structural proteins NS1 and NS2) besides the capsid proteins VP1 and VP2 that is generated by alternative splicing [1,11].

FPLV and MEV are highly contagious pathogens that have been endemic in their reservoir hosts [1,3,12–19]. The CPV-like lineage emerged in the late 1970s as a new pathogen following the occurrence of host mutations that allowed binding of the CPV-2 virus capsid proteins to

**Funding:** We thank the Agricultural Science and Technology Innovation Program (ASTIP-IAS15), the National Key Research and Development Program of China and Pró-reitoria de pesquisa e pós-graduação da UFPA (Progep #02/2019).

**Competing interests:** All the authors have declared that no competing interests exist.

the canine cellular transferrin receptor. [15,17–22]. In addition, key mutations in CVP-2 provided additional selective advantages to some CPV strains enabling them able to spread to domestic dog populations [12,15,23]. It has been suggested that both CPV and MEV are FPLV mutations that have adapted to different host animals [1,3,4,12]. Particularly, amino acids at positions 93 and 323 of the FPLV and CPV *Vp2* gene appear to influence antigenicity, and result in differences in the viral surface architecture [15,24,25]. Amino acids at positions 93, 103 and 323 of the Vp2 are critical for the stable replication of CPV in canine-derived cells, while the amino acids present at positions 80, 564, 568 are critical for FPLV replication in feline host [12,13,15,21].

In addition, some wild species such as the raccoon are important reservoir of multiple lineages of protoparvoviruses [13,15,22,23]. In fact, sporadic infections of species other than raccoon or mink have been increasingly reported over time. For example, there are cases of protoparvovirus infecting the giant panda, monkey and civets [3,4,7]. The impressive ability of protoparvovirus to infect different hosts and evade the immune responses by acquisition of mutations in their capsid protein make this virus a serious threat to wild and domestic animals [13,20,21,24–28].

FPLV has a single-stranded DNA genome; the full-length of the genome is approximately 5123 nt and contains two open reading frames (ORFs). The parvovirus genome has hairpin structures at both ends of its genome: 3-terminal Y-type structure and 5-terminal U-shaped structure [29]. These features of the genome make it challenging to amplify the full-length genome of parvovirus.

The diversity of carnivore protoparvoviruses has been described in many studies and seems to be represented by a great variety of strains and recombinant forms [27,30–32]. Conversely, there are few studies describing the heterogeneity of FPLV strains and its relevance to the maintenance of these viruses in distinct hosts [13]. Particularly, the occurrence of multiple FPLV lineages and how they contribute to the dissemination of this virus among distinct species has been explored in only a few studies [1–7,33]. Here we fully characterized two FPLV strains isolated from domestic cats presenting with gastroenteritis. Genome-wide analysis show that these FPLV isolates are distinguished from other FPLV isolates currently circulating in China by differences in several nucleotides and coding for unique amino acids in the *Vp2* and *Ns1* gene.

## 2. Materials and methods

### 2.1 Fecal samples of domestic cats

Fecal specimens were obtained by a collaborative program between the Institute of Animal Sciences and private clinics which is aimed at surveying animal infectious diseases in China. All samples were obtained with the owners consent and were collected following the recommendations of the Chinese legislation on animal protection and the guidelines of the Chinese policy and practices in veterinary medicine. Fecal samples were collected from two domestic cats that had clinical signs of suspected parvovirus infection in Hebei Province of China. They showed depression, fever, intense haemorrhagic vomiting, diarrhea, and severe leukopenia.

### 2.2 Sample treatment and confirmation of FPLV

Samples were placed into collection tubes (Yocon Bio. Co. Ltd., Beijing) and the colloidal immunochromatographic (IC) test strip in the BioNote Rapid Test Kit (BioNote. Inc. 22, Samsung1ro 4gil, Hwaseong-si, Gyeonggi-do, 18449, Republic of Korea) was used to determine whether FPLV was present. The fecal samples were mixed with PBS and the filtered supernatants were added to colloidal gold test strips at room temperature, according to the manufacturer's instructions.

Total viral DNA was extracted from fecal samples separately using the Aidlab Virus DNA Kit (Beijing Aidlab Biotech Company, Beijing, China), according to the manufacturer's instructions. The extracted DNA was used for detection of FPLV by polymerase chain reaction (PCR) using a primer pair (forward: 5'–ACAAGATAAAAGACGTGGTGTAACTCA–3' and reverse: 5'–AATGGGAAATACAGACTATAT–3') [33]. The PCR assay was performed in a 20 μL reaction volume that included 2.0 μL of template, 1.0 μL each of VP2-F and VP2-R, 2×Ex Taq mix (TaKaRa Biotechnology Co., Ltd. Nojihigashi 7-4-38, Kusatsu, Shiga, Japan 525–0058), and sufficient ddH2O to increase the volume to 20 μL. Amplification was carried out as follows: one cycle at 95°C for 5 min; followed by 35 cycles at 95°C for 50 s, 55°C for 30 s, and 72°C for 30 s; and a final extension at 72°C for 10 min. Amplicons were detected by electrophoresing 10 μL aliquots in 1.0% agarose gels in 1× TAE [40 mM Tris-acetate (pH 8.0), 1 mM EDTA].

## 2.3 Cell cultivation and virus isolation

Filtered and diluted samples were immersed in serum-free DMEM medium (DMEM, Gibco, 8717 Grovemont Cir, Gaithersburg, MD 20877, USA), and were clarified by centrifugation at 12,000 rpm/min for 10 min. The supernatant was filtered through a 0.22 μm Millipore filter (Millipore, Bedford, MA, U.S.A.) for virus isolation in Crandell feline kidney (CRFKATCC® CCL-94™) cells which contained DMEM supplemented with 2% fetal bovine serum at 37°C under 5% CO2. The virus was harvested after three rounds of plaque purification in the primary CRFK cell culture. Subsequently, the culture medium was frozen at -80°C and submitted to further passages following the same procedure after freezing and thawing for three cycles. The culture supernatants were harvested when the typical cytopathic effect (CPE) was observed in 80% of the cells.

## 2.4 Immunofluorescence assay

CRFK cells were seeded in 96-well plates, and the isolated FPLV strains were added and incubated for 36 h at 37°C under 5% $CO_2$. The DMEM medium was removed, the cells washed with phosphate buffered saline (PBST, Fisher Scientific, Loughborough, UK) carefully for three cycles. The treated cells were fixed in formaldehyde for 30 min. The primary antibody was added and incubated for 2 h, and after washing in PBST for three cycles the cells were incubated with secondary antibody for 1 h. Then the wells were washed three times by PBST, and incubated with fluorescein isothiocyanate-FITC (Fisher Scientific, Loughborough, UK) or the fluorescent stain DAPI (Fisher Scientific, Loughborough, UK) for 15 min. Finally, the plates were analyzed using a fluorescence inverted microscope. Negative controls were also prepared by the identical protocol.

## 2.5 Hemagglutinin test

The hemagglutination test was carried out using 1% red blood cells. The treated V-type microplate was stored at 4°C for 1–2 h. A positive result was defined as at least 50% red blood cells being hemagglutination positive.

## 2.6 DNA purification and sequencing

DNA was extracted from the virus in cell culture with the Aidlab Virus DNA Kit (Beijing Aidlab Biotech Company, Beijing, China), according to the manufacturer's protocol. DNA extracts were used in conventional PCR assays using the Takara LA Taq kit (TaKaRa Biotechnology Co., Ltd. Nojihigashi 7-4-38, Kusatsu, Shiga, Japan 525–0058) to amplify the genomic

fragments of FPLV. The amplification was generally achieved by means of 35 cycles of denaturation at 95˚C for50 s, annealing at 55˚C for 30 s and polymerization at 72˚C for 10 min. After electrophoresis in a 1.0% agarose gel and ethidium bromide staining, the PCR products were excised from the gel and purified by a commercial kit (Sangon Biotech, 695 Xiangmin Rd, Songjiang Qu, Shanghai Shi, China). The PCR products were cloned into pMD18-T vector (TaKaRa Biotechnology Co., Ltd. Japan). Amplified DNA was purified and then sequenced using the BigDye Terminator kit, version 3.1 (Applied Biosystems/Perkin Elmer, Foster City, CA). The samples were subjected to electrophoresis on an ABI 3130 genetic analyzer, and the sequencing data were examined using the ABI Sequencing Analysis Software). The molecular characteristics and homology of nucleotide sequences of the nearly entire viral genome were assembled and the deduced amino acids were analyzed using the MegAlign tool included in the Lasergene software package (DNASTAR, Inc.3801 Regent St.Madison, WI 53,705 USA).

## 2.7 Sequence analysis

Initially we used BLAST suite (blastn and blastx) to identify viral sequences through their similarity to annotated genomes in Genbank. Based on the best hits of blastn search 56 near-full length genomes were chosen for the following analyses. Based on the best hits of blastn search complete genomes of carnivorous parvoviruses were chosen for the analyses. These genomes were then aligned using Clustal X software [34]. Subsequently, a phylogenetic tree was constructed by the Maximum Composite Likelihood (MCL) approach assuming Hasegawa-Kishino-Yano model plus a discrete Gamma distribution (with five categories and an estimated alpha parameter = 3.1840) and the rate of invariable sites of 54.73%. All phylogenetic analyses and tree edition were conducted using MEGAX software [35].

## 2.8 Detection of recombination

To determine if our FPLV strains were free of recombination, we used the software RDP v.4 [36], which utilizes a collection of methods, an excellent and detailed explanation of each method implemented in the RDP program can be found in the user's manual (http://darwin. uvigo.es/rdp/rdp.html). Initially, we used all methods implemented in the RDP and the default parameters; these parameters were then optimized to avoid detection of false positive recombinations. In addition, window sizes of 20 to 150 as well as Bonferroni correction with p-values of 0.05 and 0.001 were utilized.

## 2.9 Amino acid ancestral reconstruction

Ancestral states were inferred using the Maximum Likelihood method under the Whelan and Goldman model [37]. The tree shows a set of possible amino acids (states) at each ancestral node based on their inferred likelihood at site 1. The initial tree was inferred using this method. The rates among sites were treated as a Gamma distribution using 5 Gamma Categories (Gamma Distribution option). The coding data were translated assuming a Standard genetic code table. All positions containing gaps and missing data were eliminated. Ancestral reconstructions and evolutionary distances analyses were conducted in MEGAX.

## 3. Results

### 3.1. Confirmation of *FPLV* in the fecal samples by rapid test and PCR

Fecal samples were diluted and then prepared for screening using both the colloidal gold strip test and PCR. One example of a positive sample is shown in Fig 1. The left panels of the figure show the result of the colloidal gold test strip, the upper panel is a negative control sample (Fig

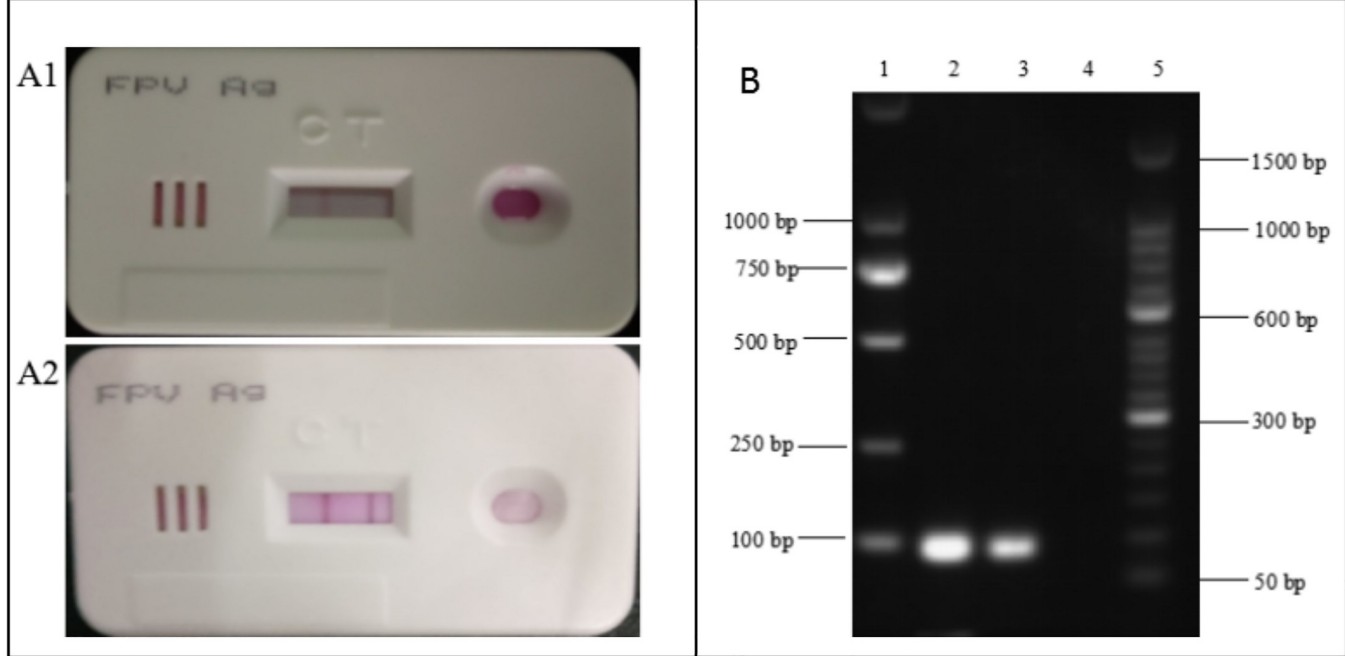

**Fig 1. FPLV detection by colloidal gold test strip and PCR assay.** A Detection of the FPLV antigen by colloidal gold test strip. A1 negative control, A2 positive sample. B Detection of FPLV in samples by PCR. Lanes 1 and 5 DNA marker; Lane 2 positive control; Lane 3 positive sample Lane 4 negative control.

1A1) and the lower panel is a positive sample (Fig 1A2). Samples were also screened using a PCR assay. The right panel in the figure shows a positive PCR control (lane 2), one positive sample (lane 3) and a negative PCR control (lane 4). The PCR positive samples presented with a band at 83 base pairs (bp), according to the expected amplicon size when using primers described previously [33].

### 3.2. Morphological changes in *FPLV*-infected cells and identification of *FPLV* by HA and IFA

CRFK cells were incubated with the FPLV strains and a typical cytopathic effect of the virus was observed in after the 3rd passage. FPLV-infected cells showed extensive regions of cell detachment (Panel A in Fig 2).

The hemagglutination test (HA) was carried out by reacting the virus with 1% porcine red blood cells at 4°C for 1–2 h. The ability to agglutinate the porcine red blood cells is consistent with the characteristics of FPLV. FPLV strains were also subjected to three rounds of plaque purification in CRFK cells. Afterwards, an indirect immunofluorescence assay (IFA) of our FPLV strains in CRFK cells was performed The positive group exhibits specific green (FITC) and blue (DAPI) fluorescences while the control group had no fluorescence (Fig 3, panel A).

### 3.3. Analysis of the nearly complete *FPLV* genomes

The genomes of the FPLV strains were fully characterized. These strains were isolated between 2014 and 2015 in domestic cats from Hubei, a province in China. Sequences generated in this study, denoted herein as LSJ-2014 (Genbank number: MH165481) and CSJ-2015 (Genbank number: MH165482), have 99% nucleotide identity and 100% of genome coverage to some FPLV, MEV and CPV strains from Genbank (FVP: M38246, KX900570, KP769859, MEV: D00765 and CPV: M38245). Both isolates LSJ-2014 and CSJ-2015 have genomes of 5118 nt in

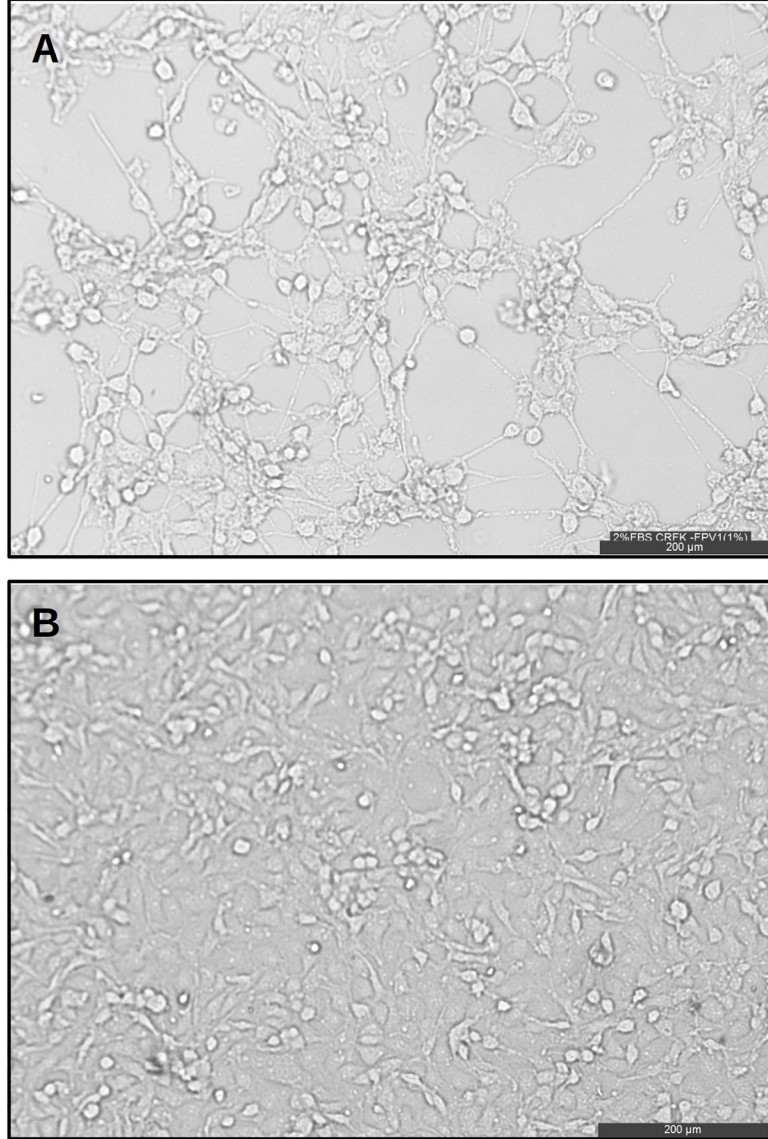

**Fig 2. Cytopathic effect of feline panleukopenia virus on CRFK cells.** Panel A shows the cytopathic effect induced by FPLV in CRFK cells visualized by inverted microscopy at magnification of 200 μm 72 hours post infection. Panel B is the negative control of uninfected CRFK cells.

length and contain the ORFs: NS1 that encodes a protein of 668 amino acids that starts at position 267 and ends at position 2270 of the genome; VP1 that encodes a protein of 727 amino acids generated by the alternative splicing of the fragment located at positions 2280–2308 plus the fragment corresponding to positions 2381–4535; the VP2 that encodes a protein of 668 amino acids that starts at position 2781 and ends at position 4535 of the genome. Table 1 shows differences in the amino acids of the NS1, VP1 and VP2 proteins between the isolates LSJ-2014 and CSJ-2015.

### 3.4. Analysis of nearly completed genomes of carnivorous parvoviruses

For comparative purposes full length genomes of FPLV, CPV and MEV were analyzed. The maximum likelihood phylogenetic tree inferred with most of the complete genomes of CPV,

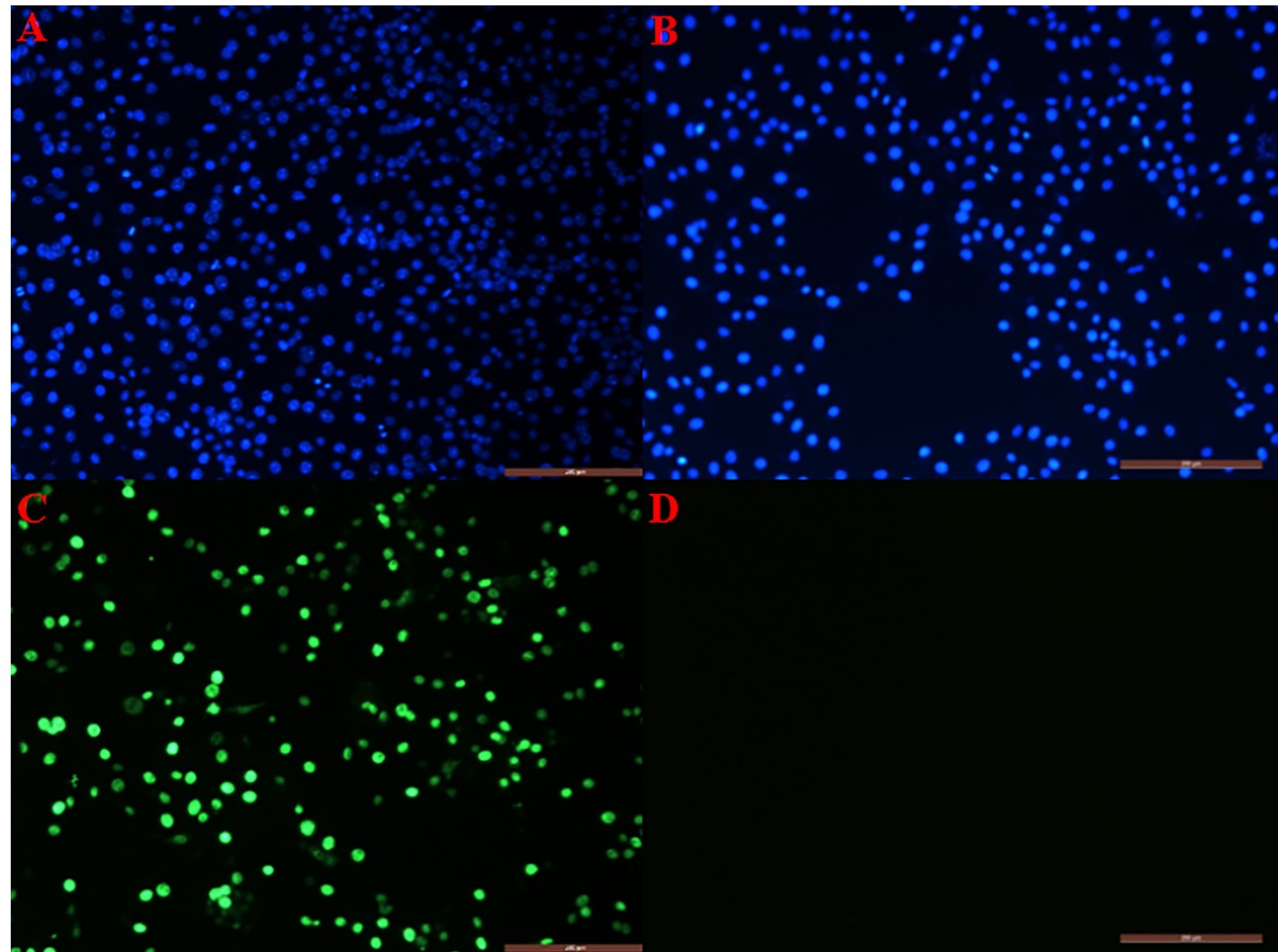

**Fig 3. Indirect immunofluorescence assay of FPLV in CRFK cells.** Immunofluorescence staining of CRFK cells after infection with FPLV. Panels A and B indicate nucleous stained with DAPI. C) Mock-infected controls. Uninfected CRFK cells served as a negative control (D). A green color indicates positive Fluorescein isothiocyanate (FITC) staining and blue indicated the nuclear stained with DAPI of FPLV infected cells. Scaled indicate of 200 μm of magnification.

FPLV and MEV showed that CPV forms a well separated clade from FVP and MEV sequences (Fig 4). Particularly, the CPV clade has clusters of sequences that represent strains from some geographical regions such as CPV2c from Uruguay. It is also seen that CPV infecting raccoons

**Table 1. Differences in the amino acid composition between LSJ-2014 and CSJ-2015.**

| FPLV Isolate | Protein | |
|---|---|---|
| | NS1 | VP1 |
| LSJ-2014 | 11T, 93N, 247Y, 370I | 35S, 50L, 71T |
| CSJ-2015 | 11M,93D, 247H,370M | 35F, 50F, 71N |

IUPAC code: A = Alanine; C = Cysteine; D = Aspartic Acid; E = Glutamic Acid; F = Phenylalanine; G = Glycine; H = Histidine; I = Isoleucine; K = Lysine; L = Leucine; M = Methionine;N = Asparagine; P = Proline; Q = Glutamine; R = Arginine; S = Serine; T = Threonine; V = Valine; W = Tryptophan; Y = Tyrosine

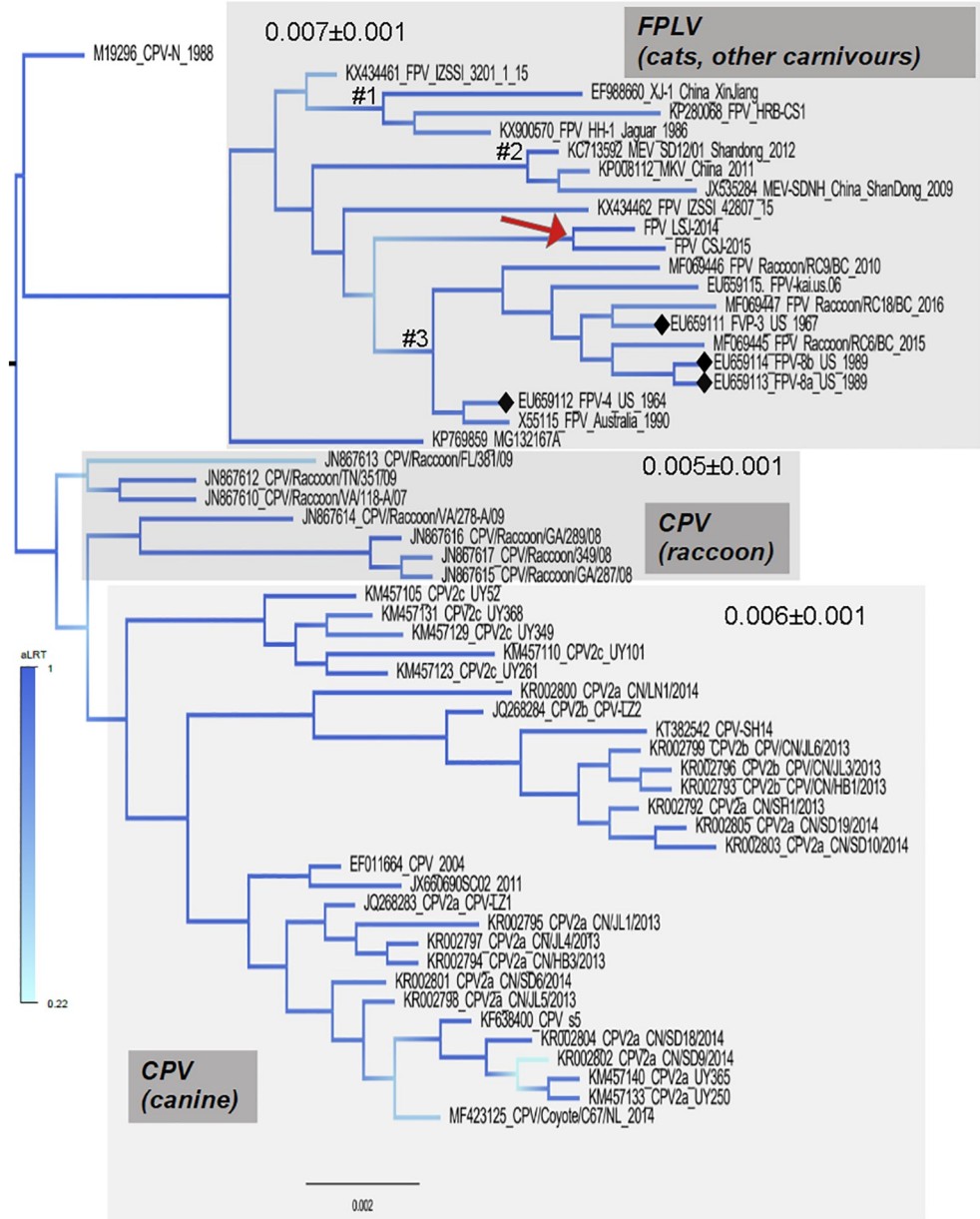

**Fig 4. Phylogenetic tree constructed using near-full genomes of carnivorous parvoviruses.** The tree was inferred using a maximum likelihood (ML) approach assuming HKY model plus a discrete Gamma correction and the rate of invariable sites. The FPLV strains isolated in this study are indicated by red arrow in the tree. Branch support is indicated by a colored scale and were obtained by the approximate likelihood ratio test (aLRT) where one is the highest score. Black diamonds indicate the position of the historical FPLV strains; those that were isolated prior or near the spread of CPV in domestic dogs. Some clusters

of FPLV or MEV that are discussed in the manuscript are indicated by the symbol #. Values inside the gray areas indicate the genetic distances of each clade. The Genbank identification of all sequences used in the analyses are listed in Material and Methods. Phylogenetic analyses and estimation of genetic distances were conducted with MEGAX software.

in the USA that were sampled in 2007 and 2009 cluster together at the base of the tree. Conversely, FPLV and MEV clustered in the same phylogroup which is expected because these viruses are related to some extent. FPLV strains generated in this study (LSJ-2014 and CSJ-2015) are indicated in the tree by a gray arrow. They are located at the base of the cluster identified by #3 which is formed by strains from raccoons and historical strains (*i.e.*, those isolated in 1960s and early 1980s before the virus adapt and spread in domestic dogs) from wild and domestic cats. Interestingly, FPLV strains isolated in distinct species intermingled in the tree, with the exception of a cluster of MEV from China (identified by #2 in the tree of Fig 4). For example, the cluster #3 formed by FPLV from raccoons (isolated between 2010 and 2016) also includes FPLV strains from domestic cats isolated between 1967 and 2006. Besides the cluster formed by MEV from China (#2) there is also another cluster containing strains from domestic cats and one strains from jaguars (identified by #2 in the tree of Fig 4).

### 3.5. NS1, VP1 and VP2 trees of carnivorous parvoviruses

We were interested to know if the topology of trees constructed using the main parvoviral genes would match to the pattern of relatedness observed in the genome tree. Therefore we used the same set of complete genomes and partitioned the alignment into tree segments corresponding to the genes *Ns1*, *Vp1* and *Vp2*, and the partitions were used to infer maximum likelihood trees (Fig 5). These trees showed that the same strains were included in three main phylogroups, designated #1, #2 and #3, the exceptions were strains LSJ-2014, CSJ-2015 (indicated by a blue arrow in the trees of Fig 5), KP769859 (detected in 2016 in a domestic cat in Belgium), KX434461 and KX434462 (indicated by green arrows in the trees of the Fig 5) both detected in Italy in 2015 from domestic cats.

### 3.6. Ancestral reconstruction of amino acids of VP2 in carnivorous parvoviruses

We explored if the conflicting location of LSJ-2014, CSJ-2015 in the maximum likelihood trees was related to some biological property of these sequences or an artifact owing to the small number of sequences included in the previous phylogenetic analysis. For the analysis we used 148 VP2 sequences of FPLV and MEV selected from the Genbank (Ids included in the S1 Table). These sequences were selected basically by the absence of recombination. Recombinant strains need to be removed because recombination is a significant bias to the inferences of trees based on maximum likelihood or Bayesian approaches [31]. The presence of positive selection was also measured in the alignments of VP2 in order to contemplate the effect of positive selection on the topology of the trees. The VP2 tree and amino acid reconstruction are shown in Fig 6. This tree indicated that there are clusters formed exclusively by MEV isolates (identified in the tree by blue circles). A different evolutionary scenario was observed in the FPLV isolates: strains detected in raccoon, fox, tiger, lion, wildcat, domestic cat and puma are intermingled in the VP2 tree (indicated by circles and names with different colors). Phylogroups are generally formed by strains of distinct species and different regions. The rectangular gray areas in the tree indicate mutations that occurred in some lineages.

The tree also shows that there are no species-specific FPLV strains. However, there are some clusters formed by regional strains. For instance, clade # 1 of the tree is composed mainly

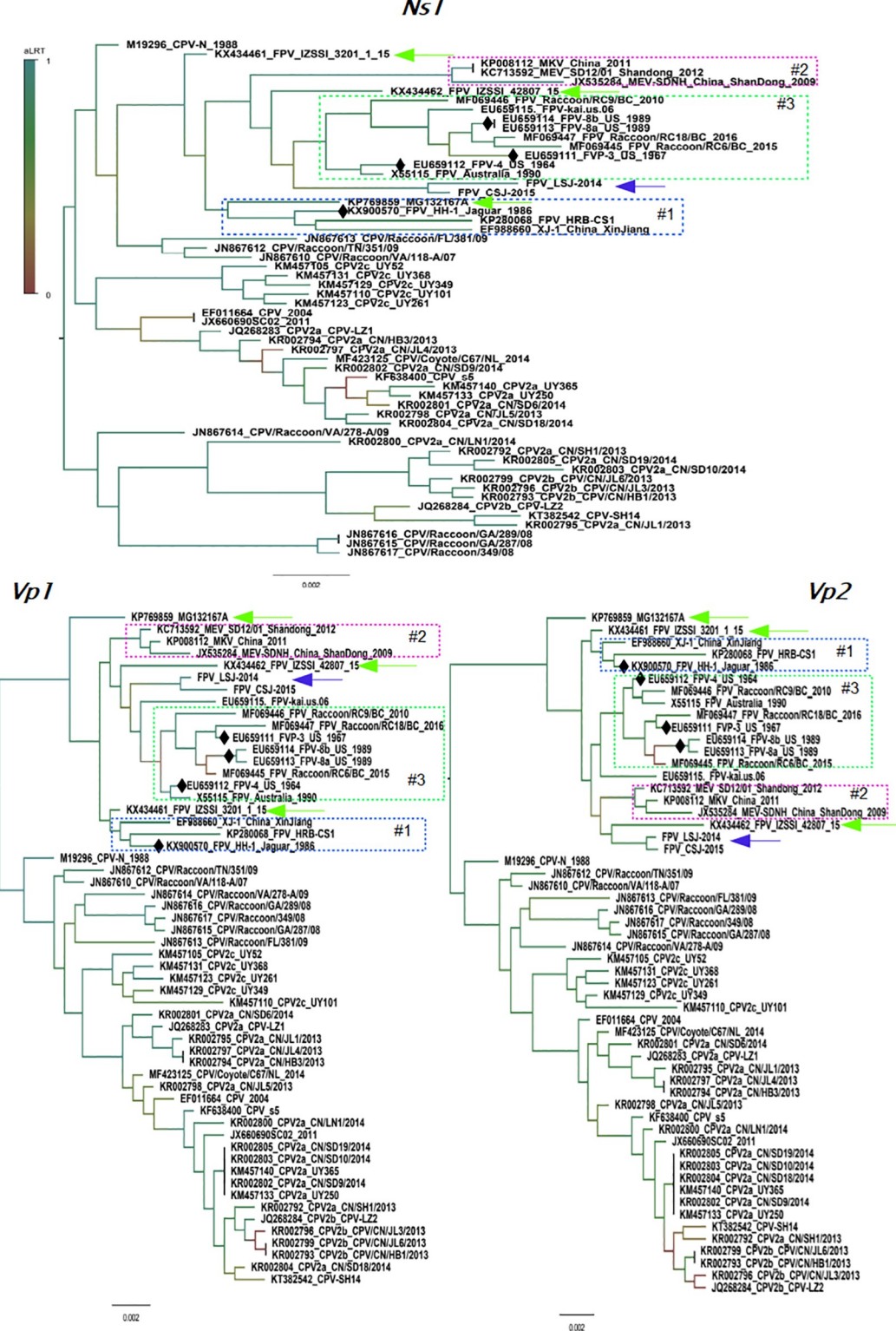

**Fig 5. Phylogenetic tree constructed using NS1, VP1 and VP2 of carnivorous parvoviruses.** Partitions corresponding to the genes *Ns1*, *Vp1* and *Vp2* were used to construct trees and show the clustering pattern of sequences. The upper panel shows the tree inferred using the partitions corresponding to the *Ns1* gene of parvoviruses (2004 nucleotides long). The lower panels in the figure show the trees inferred using the *Vp1* gene region with 2182 nucleotides and the *Vp2* gene with 1753 nucleotides. All trees were inferred using a maximum likelihood (ML) approach assuming the HKY model plus a

discrete Gamma correction and the rate of invariable sites. The FPLV strains isolated in this study are indicated by blue arrows in the tree. Strains KP769859, KX434461 and KX434462 are indicated by green arrows in the trees. Branch support are indicated in a colored scale and were obtained by the approximate likelihood ratio test (aLRT) where one is the highest score. Black diamonds indicate the position of historical FPLV strains, those that were isolated prior or near the spread of CPV in domestic dogs. Some clusters of FPLV or MEV discussed in the manuscript are indicated by numbers. The Genbank identification of all sequences used in the analyses are listed in Material and Methods. Phylogenetic analyses and estimation of genetic distances were conducted in MEGAX software.

of FPLV strains from domestic cats, one lion and one tiger. These sequences were isolated in Europe (Portugal and Italy). Another example of regional clustering is group #2 composed of strains from domestic cats and also a unique group of MEV strains (indicated by a blue dot). All strains of this group are from China. Most clusters have particular sets of amino acids that make them distinct from other strains. Mutations at cluster #1, #2 and #5 likely are local adaptations because they are composed of regional strains. Cluster #6 is composed of strains from Korea but no peculiar mutations was detected in the ancestral reconstruction.

## 4. Discussion

Several amino acid residues in the VP2 protein influence the antigenicity and host range of both CPV-2 and FPLV [12,18]. FPLV and CPV showed a completely different pattern of evolution, although they are highly related at the genetic level. FPLV has evolved mainly by random genetic drift and multiple strains likely use feral cats and other carnivores as reservoirs [12,14,15,17–19,21,26,38]. However, CPV evolution, which leads to the emergence of new antigenic variants (2a and 2b) and expansion of the host range, was partly driven by positive selection [17–19,27,28]. FPLV has not experienced major changes in antigenic and biological properties since its first identification in 1920 [19]. The major antigenic site of parvoviruses is on the VP2 protein, and mutations in the VP2 protein at resides 80(K), 93 (K), 103(V), 323(D), 564(N) and 568(A) will affect the antigenic characteristics and host range [3,12,13,15,17,29].

Our phylogenetic analysis of near-full length genomes of FPLV showed that historical strains (those isolated in the 1970s and 1980s) usually are not located at the base of the phylogroups. In addition, historical strains are evenly mixed with recently isolated strains. This lack of temporal structure of the PFV suggests the absence of key strains that could provide more information regarding the spread of this virus in carnivorous species. The maximum likelihood trees also showed many events of cross-infections between FPLV and MEV. Additionally, the phylogenetic analysis and the ancestral reconstruction showed that FPLV is able to infect distinct carnivores with little changes in the *Vp2* gene. The evolutionary history of CPV comprises highly host-adapted strains [2,21,27]. FPLV, on the other hand, seems to be less specialized. Particularly, FPLV lineages infecting domestic cats and wild cats are nearly identical. It is interesting to mention that certain FPLV strain which are less pathogenic in domestic cats can be lethal to big felids such as tigers and lions [32,39,40].

In summary, our analysis indicated that some FPLV and MEV lineages likely evolve in local populations merely by changes in nucleotide composition instead of protein mutations. Interestingly, our FPLV isolates (LSJ-2014 and CSJ-2015) and MERV from China diverged from lineages containing strains used in some vaccines. Thus it is quite reasonable to suggest they may have emerged as evasion strains due to selective pressures of immune surveillance. This feature is in concert with a previous report indicating that FPLV may evolve primarily by genetic drift. The near-complete genome of FPLV strain in this study is of great potential value to further exploring antigenic variation and development of a genetically engineered vaccine.

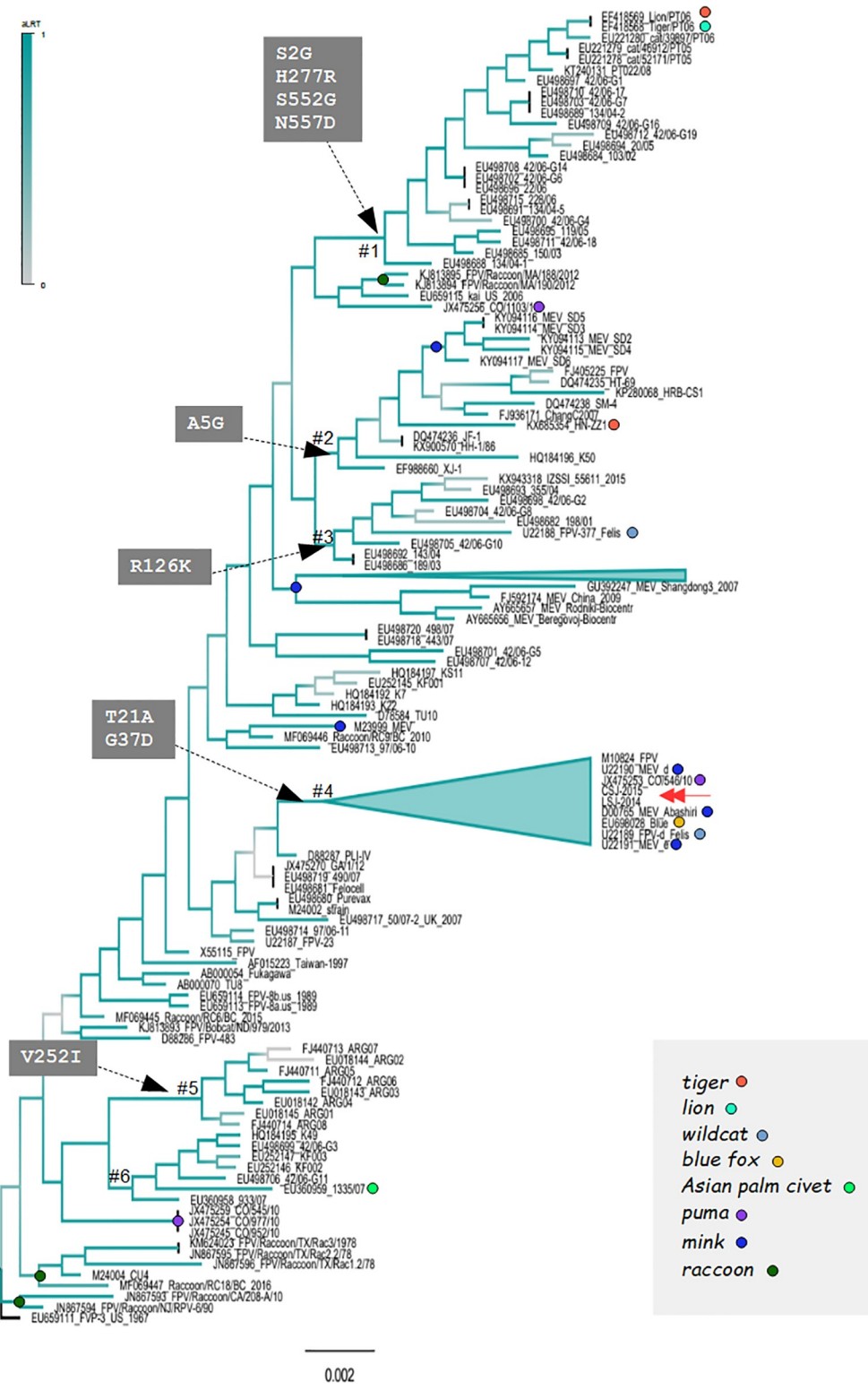

**Fig 6. Ancestral reconstruction of amino acids of VP2.** The tree was inferred using a maximum likelihood (ML) approach assuming the HKY model plus a discrete Gamma correction and the rate of invariable sites. The FPLV strains isolated in the present study are indicated by red arrow in the tree. Branch support are indicated by a color scale and were obtained by the approximate likelihood ratio test (aLRT) where one is the highest score. Some clusters of FPLV or MEV that are discussed in the manuscript are indicated by hashtag symbols. Circles with distinct colors

indicate strains isolated in different species according to the color code in the figures in the right portion of the tree. Grey rectangles indicate amino acid changes that occurred in some clusters, numbered rectangles indicate the position in the VP2 protein. The letters indicate the one letter amino acid code as follows: A = Alanine; C = Cysteine; D = Aspartic Acid; E = Glutamic Acid; F = Phenylalanine; G = Glycine; H = Histidine; I = Isoleucine; K = Lysine; L = Leucine; M = Methionine;N = Asparagine; P = Proline; Q = Glutamine; R = Arginine; S = Serine; T = Threonine; V = Valine; W = Tryptophan; Y = Tyrosine. The Genbank identification of all sequences used in the analyses are listed in Material and Methods. Sequences used to infer the VP2 tree were absent from any signal of recombination according to the analysis based on distinct methods as implemented in the RDP v.4 software.

## Supporting information

**S1 Raw Images.**
(PDF)

**S1 Table. Identification of FLPV sequences used in the phylogenetic analyses.**
(PDF)

## Author Contributions

**Conceptualization:** Élcio Leal, Lijun Shi, Jinxiang Li, Shangjin Cui.

**Data curation:** Lin Liang.

**Formal analysis:** Élcio Leal, Ruiying Liang, Fabiola Villanova.

**Funding acquisition:** Élcio Leal, Jinxiang Li, Shangjin Cui.

**Investigation:** Lijun Shi, Shangjin Cui.

**Methodology:** Ruiying Liang, Qi Liu, Fabiola Villanova, Lin Liang.

**Project administration:** Shangjin Cui.

**Supervision:** Élcio Leal, Shangjin Cui.

**Writing – original draft:** Élcio Leal, Shangjin Cui.

**Writing – review & editing:** Steven S. Witkin.

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
