## [Decision Letter · Decision Letter 0]

12 Sep 2019

PONE-D-19-23818

Regional adaptations and parallel mutations in Feline panleukopenia virus strains from China revealed by nearly-full length genome analysis

PLOS ONE

Dear Dr. Leal,

Thank you for submitting your manuscript to PLOS ONE. After careful consideration, we feel that it has merit but does not fully meet PLOS ONE’s publication criteria as it currently stands. Therefore, we invite you to submit a revised version of the manuscript that addresses the points raised during the review process. In particular, please provide background information of the prototype virus strain for comparison and any information regarding the immune status of the animals from which virus strains were isolated.  

We would appreciate receiving your revised manuscript by Oct 27 2019 11:59PM. To enhance the reproducibility of your results, we recommend that if applicable you deposit your laboratory protocols in protocols.io, where a protocol can be assigned its own identifier (DOI) such that it can be cited independently in the future. For instructions see: http://journals.plos.org/plosone/s/submission-guidelines#loc-laboratory-protocols

We look forward to receiving your revised manuscript.

Kind regards,

Jianming Qiu, Ph.D.

Academic Editor

PLOS ONE

Journal Requirements:

3.  We note that Figure(s) in your submission contain copyrighted images. All PLOS content is published under the Creative Commons Attribution License (CC BY 4.0), which means that the manuscript, images, and Supporting Information files will be freely available online, and any third party is permitted to access, download, copy, distribute, and use these materials in any way, even commercially, with proper attribution. For more information, see our copyright guidelines: http://journals.plos.org/plosone/s/licenses-and-copyright.

1.    You may seek permission from the original copyright holder of Figure(s) [#] to publish the content specifically under the CC BY 4.0 license.

Additional Editor Comments (if provided):

Reviewers' comments:

Reviewer's Responses to Questions

**Comments to the Author**

1. Is the manuscript technically sound, and do the data support the conclusions?

Reviewer #1: Yes

Reviewer #2: Yes

2. Has the statistical analysis been performed appropriately and rigorously? 

Reviewer #1: N/A

Reviewer #2: N/A

3. Have the authors made all data underlying the findings in their manuscript fully available?

Reviewer #1: Yes

Reviewer #2: Yes

4. Is the manuscript presented in an intelligible fashion and written in standard English?

Reviewer #1: Yes

Reviewer #2: Yes

5. Review Comments to the Author

Reviewer #1: In the present study, Leal and colleagues isolated two Feline panleukopenia viruses from domestic cats. They performed the phylogenetic analysis of the viral genome and evolutionary analysis of the VP2 gene. They found the isolated stains are close to historical strains of FPLV/MEV and the VP2 gene is characterized by local adaptation. This study supplies more information about the genomics and evolutionary information of FPLV.

Major comments to the authors.

1. The authors compared the isolated two FPLV strains with historical strains. But few information was introduced about these historical strains. More background information about historical strains should be included.

2. In the Ancestral reconstruction section, P13, line 271-308, the authors claimed “the conflicting location of LSJ-2014, CSJ-2015 in the maximum likelihood trees”, please explain more about this conflicting location of LSJ-2014, CSJ-2015 in the maximum likelihood trees. VP1 only contains 59 more amino acids (177nt) than VP2. Why the phylogenetic tree of VP1 and VP2 is so different (conflicting)? the authors showed some mutations in the roots of #1, #2, #3, #4, and #5 in figure 6, does that mean all viruses in this branch contain the same mutations?

3. The authors claimed that the isolated FPLV may emerged as evasion strains due to selective pressures of immune surveillance. Do you have any information about the immune situation of the cats which the two FPLV isolated from?

Minor comments to the authors.

1. P8, line 47, when you introduce amino acids, it is better use VP2 protein but not Vp2 gene.

2. P8, line 49-50, typos, “presente”, “posiions”

3. P9, line 92, not sure why include (Table S1) here.

4. P9, line 112-115, the buffer is PBST but not PBS? What is the primary and secondary antibodies used for IFA? What fluorescent dye was conjugated in the second antibody? Typo, “aded” in line 113. “wells” in line 114 is better use “cells”.

5. P10, line 128-137, the primers used for amplifying the viral genome should be included. It is not rigorous to claim “entire viral genome” in line 137.

6. P11, line 179, Lane 2 is for positive control, lane 3 is for positive sample and lane 4 is negative control.

7. P11, line 197 and figure 3, figure 3A is showing cells infected by virus, and figure 3B is negative control. DAPI staining should be included in the image.

8. P12, line 207-210, it is not rigorous to call NS1, VP1, and VP2 polyprotein.

9. P12, line 215-218, some amino acids were not included in Table 1. Why B is representing Aspartic acid? Y which showed in the table is not included.

10. P12, line 227 and figure 4, why BJ-04 and BJ-05 were used for isolated viruses? Please consistent in the manuscript.

11. P12, line 228, a red arrow was used in the figure.

12. P13, line 255-256, KP769859, KX434461, and KX434462 were not indicated by arrows.

13. P13, line 264-265, no values showed in the figure.

14. P14, line 285, vaccine strains did not show in the figure.

15. P14-15, line 283-301, the authors claim that “Phylogroups are generally formed by strains of distinct species” which is conflict with the statement of “The tree also shows that there are no species-specific FPLV strains”. Please double check. Line 290-291, no values showed in the figure.

Reviewer #2: The manuscript entitled, “Regional adaptations and parallel mutations in Feline panleukopenia virus strains from China revealed by nearly-full length genome analysis” by Leal E. et al. identified two FPLV strains in domestic cats from Hubei, china. The authors show that the two strains are closely related to FPLV/CPV isolates from 1960’s. They also demonstrate that the two strains have diverged in Vp gene sequence and therefore possibly evade anti-FPLV immune reponse by the current vaccine strains.

The authors need to address the following concerns to further substantiate their claims and make the required changes before the manuscript be considered suitable for publication in Plos One.

1. It is mentioned in the materials and methods for Immunofluorescence assay that the cells were DAPI stained for 15 mins. However, in Fig. 3, there are no DAPI stained images at all. The authors better show the bright field images of the cells infected/uninfected or cells be stained with DAPI. Also, are these images taken at 200X, as mentioned in Line 199?

2. It is quite difficult to read the text lines in Fig. 4. Authors must change these text lines to make it legible.

3. There are several grammatical mistakes, typos and other inconsistencies in the text. The authors need to pay attention to these mistakes and make the necessary changes. I have mentioned many of them here:

Line 40… “Ns2” here is a protein, not a gene and should be written as “NS2”.

Line 49,50… “presente at posiions” should be as “present at positions”.

Line 53… “increasing” should be “increasingly”.

Line 66… Genome-wide “analyze” as Genome-wide “analysis”.

Line 194… “exhibites”, should be “exhibits”

Line 225… “Seem” should be “seen”.

Line 293… should be “according”.

6. PLOS authors have the option to publish the peer review history of their article (what does this mean?). If published, this will include your full peer review and any attached files.

Reviewer #1: No

Reviewer #2: No

---

## [Author Response · Author response to Decision Letter 0]

20 Dec 2019

COMMENTS:

Reviewer #1: In the present study, Leal and colleagues isolated two Feline panleukopenia viruses from domestic cats. They performed the phylogenetic analysis of the viral genome and evolutionary analysis of the VP2 gene. They found the isolated stains are close to historical strains of FPLV/MEV and the VP2 gene is characterized by local adaptation. This study supplies more information about the genomics and evolutionary information of FPLV.

Major comments to the authors.

1. The authors compared the isolated two FPLV strains with historical strains. But few information was introduced about these historical strains. More background information about historical strains should be included.

Resp: Historical samples are those isolated in cats prior the spread of the FPLV to the dogs which occurred before 1980s (molecular clock estimates that the host transmission occurred between 1970-1978). After 1980 the virus adapted to dogs and was named CPV-2 and soon a new variant (named CPV-2a) replaced CPV-2. In the original manuscript a brief discussion about these strains was included (Lines 316-320) we also included some explanation in the lines 22-224.

2. In the Ancestral reconstruction section, P13, line 271-308, the authors claimed “the conflicting location of LSJ-2014, CSJ-2015 in the maximum likelihood trees”, please explain more about this conflicting location of LSJ-2014, CSJ-2015 in the maximum likelihood trees.VP1 only contains 59 more amino acids (177nt) than VP2.

Resp: Since phylogenetic trees can rotate vertically and this has no meaning to the topology then Vp1, Vp2 and Ns2 trees are not distinct in their topology because clade #1, #2 and #3 are present in all trees. Please note that in in all trees clades #2 and #3 are related (they share a common ancestor node). Also note that strains LSJ-2014 and CSL-2015 are closely related with the clade #3 in the Ns1 and Vp1 tree while in the Vp2 tree these strains are closely related with the clade #2. 

Why the phylogenetic tree of VP1 and VP2 is so different (conflicting)? the authors showed some mutations in the roots of #1, #2, #3, #4, and #5 in figure 6, does that mean all viruses in this branch contain the same mutations? 

Resp: No necessarily all strains in the leafs of a tree share the same residue that was inferred in a certain node because the ancestral reconstruction relies on phylogenies and it is based on a codon model that follows an evolutionary approach. So the probability of a residue in a hypothetical ancestral protein depends not only on the frequency of all amino acids in the alignment, it also depends on the likelihood of having that residue at a certain internal node of the inferred tree.

3. The authors claimed that the isolated FPLV may emerged as evasion strains due to selective pressures of immune surveillance. Do you have any information about the immune situation of the cats which the two FPLV isolated from?

Resp: Besides depression, fever, intense haemorrhagic vomiting, diarrhea, and severe leukopenia no other information was available 

Minor comments to the authors.

1. P8, line 47, when you introduce amino acids, it is better use VP2 protein but not Vp2 gene.

Resp: we did change in the manuscript

2. P8, line 49-50, typos, “presente”, “posiions”

Resp: we did change in the manuscript

3. P9, line 92, not sure why include (Table S1) here.

Resp: “Table S1” was removed from this part of the text

4. P9, line 112-115, the buffer is PBST but not PBS? What is the primary and secondary antibodies used for IFA? What fluorescent dye was conjugated in the second antibody?

Resp: PBST is PBS plus Tween used as standard washing buffers.

In immunolabeling assays primary antibodies bind directly to the antigens and secondary antobodies bind to the Fc domains of primary antibodies. The Fc domain is constant within the same animal class. The Fc domain of secondary antibodies are labeled (usually with FITC) only one type of secondary antibody is required to bind to many types of primary antibodies, this reduces the cost by labeling.

Typo, “aded” in line 113. “wells” in line 114 is better use “cells”.

Resp: This was changed in the manuscript.

5. P10, line 128-137, the primers used for amplifying the viral genome should be included. It is not rigorous to claim

Resp: Primers were included in the section 2.2.”Sample treatment and confirmation of FPLV”

“entire viral genome” in line 137.

Resp: we change this to nearly entire genome

6. P11, line 179, Lane 2 is for positive control, lane 3 is for positive sample and lane 4 is negative control.

Resp: This was corrected in the new version of the manuscript

7. P11, line 197 and figure 3, figure 3A is showing cells infected by virus, and figure 3B is negative control. DAPI staining should be included in the image.

Resp: In the new figure 3 DAPI staining of nucleus was included

8. P12, line 207-210, it is not rigorous to call NS1, VP1, and VP2 polyprotein.

Resp: This also was corrected in the new version of the manuscript

9. P12, line 215-218, some amino acids were not included in Table 1. Why B is representing Aspartic acid? Y which showed in the table is not included.

Resp: this was a mistake, Amino acid UIPAC codes were corrected in the new manuscript.

10. P12, line 227 and figure 4, why BJ-04 and BJ-05 were used for isolated viruses? Please consistent in the manuscript.

Resp: We changed these names in the text of the manuscript and also in the figure4.

11. P12, line 228, a red arrow was used in the figure.

Resp: this was corrected in the new manuscript.

12. P13, line 255-256, KP769859, KX434461, and KX434462 were not indicated by arrows.

Resp: They are indicted by green arrows in the new version of the figure 5.

13. P13, line 264-265, no values showed in the figure.

Resp: We have changed this in the text to indicate that branch support are indicated by a color scale in the trees.

14. P14, line 285, vaccine strains did not show in the figure.

Resp: this was removed from the new version of the manuscript.

15. P14-15, line 283-301, the authors claim that “Phylogroups are generally formed by strains of distinct species” which is conflict with the statement of “The tree also shows that there are no species-specific FPLV strains”.

Resp: It is true that a same FLPV strain can infect distinct hots (for example; raccoons and cats). By this reason phylogenetic clades are composed by strains from different species. The exception is MRV that circulate in minks kept in close contacts in farms. 

Please double check. Line 290-291, no values showed in the figure.

Resp: We changed this in the new manuscript.

Reviewer #2: The manuscript entitled, “Regional adaptations and parallel mutations in Feline panleukopenia virus strains from China revealed by nearly-full length genome analysis” by Leal E. et al. identified two FPLV strains in domestic cats from Hubei, china. The authors show that the two strains are closely related to FPLV/CPV isolates from 1960’s. They also demonstrate that the two strains have diverged in Vp gene sequence and therefore possibly evade anti-FPLV immune reponse by the current vaccine strains.

The authors need to address the following concerns to further substantiate their claims and make the 

required changes before the manuscript be considered suitable for publication in Plos One.

1. It is mentioned in the materials and methods for Immunofluorescence assay that the cells were DAPI stained for 15 mins. However, in Fig. 3, there are no DAPI stained images at all. The authors better show the bright field images of the cells infected/uninfected or cells be stained with DAPI. Also, are these images taken at 200X, as mentioned in Line 199?

Resp: Out fault, we did include the image of DAPI staining in the figure 3 and also changed the magnification to 200 μm in the new manuscript. 

2. It is quite difficult to read the text lines in Fig. 4. Authors must change these text lines to make it legible.

Resp: Figure 4 was modified and the quality and font size were changed.

3. There are several grammatical mistakes, typos and other inconsistencies in the text. The authors need to pay attention to these mistakes and make the necessary changes. I have mentioned many of them here:

Line 40… “Ns2” here is a protein, not a gene and should be written as “NS2”.

Resp: This was changed in the new manuscript

Line 49,50… “presente at posiions” should be as “present at positions”.

Resp: These typos were changed in the new manuscript

Line 53… “increasing” should be “increasingly”.

Resp: This was changed in the new manuscript

Line 66… Genome-wide “analyze” as Genome-wide “analysis”.

Resp: This was changed in the new manuscript

Line 194… “exhibites”, should be “exhibits”

Resp: This was changed in the new manuscript

Line 225… “Seem” should be “seen”.

Resp: This was changed in the new manuscript

Line 293… should be “according”.

Resp: This was changed in the new manuscript

---

## [Editor Report · Decision Letter 1]

27 Dec 2019

Regional adaptations and parallel mutations in Feline panleukopenia virus strains from China revealed by nearly-full length genome analysis

PONE-D-19-23818R1

Dear Dr. Leal,

We are pleased to inform you that your manuscript has been judged scientifically suitable for publication and will be formally accepted for publication once it complies with all outstanding technical requirements.

With kind regards,

Jianming Qiu, Ph.D.

Academic Editor

PLOS ONE
---

## [Editor Report · Acceptance letter]

30 Dec 2019

PONE-D-19-23818R1 

Regional adaptations and parallel mutations in Feline panleukopenia virus strains from China revealed by nearly-full length genome analysis 

Dear Dr. Leal:

I am pleased to inform you that your manuscript has been deemed suitable for publication in PLOS ONE. Congratulations! Your manuscript is now with our production department. 

With kind regards,

on behalf of

Dr. Jianming Qiu 

Academic Editor

PLOS ONE